# A comparative driving safety study of mountainous expressway individual tunnel and tunnel group based on eye gaze behavior

**Ting Shang, Hongjiao Qi, An Huang◉\*, Tangzhi Liu**

School of Traffic & Transportation, Chongqing Jiaotong University, Chongqing, China

\* 622200950083@mails.cqjtu.edu.cn

## Abstract

The traffic environment of a tunnel group is more complex than that of a single tunnel, which increases the driving risk. The eye gaze behavior of drivers can be used to evaluate driving safety and comfort. To analyze the fixation characteristics of drivers in a single tunnel and tunnel group of mountainous expressways, an actual vehicle test is conducted. The test area has a total length of 160 km and 38 tunnels, including 8 tunnel groups and 16 single tunnels. In the test, the difference in the gaze time of five drivers between single tunnels and tunnel groups is compared. The k-means method is used to cluster driver's gaze points dynamically. Based on the Markov theory, the attributes related to gaze transfer are obtained. The results show that when tunnels are of short or medium length, there is no significant difference in the gaze time and gaze point transfer between the tunnel group and a single tunnel. In contrast, when tunnels have long or extra-long length, the repeated fixation probability and the two-step transition probability of looking back of a driver in a tunnel group are higher than those in a single tunnel. The design and management method of a single tunnel cannot be directly used, especially for extra-long tunnels located at the back of a tunnel group with a long upstream tunnel length and a short interval distance from the upstream tunnel. Therefore, it is necessary to focus on the design and management methods of tunnel groups.

## 1. Introduction

A tunnel has obvious advantages in breaking through the geographical barrier and shortening the journey. However, due to a single closed space, poor line of sight, and sharp change in brightness at the entrance and exit of a tunnel, there are problems of high traffic intensity and accident rate. With the development of expressways to remote mountainous areas, the number and scale of tunnels have increased. Currently, the largest tunnel group in the world includes 136 tunnels and is on the Xihan Expressway in Shanxi Province, China. Due to a large number of tunnels and a short distance between them, the light environment at the entrance and exit of a tunnel group changes dramatically, which puts an excessive load on a driver's vision and makes the traffic safety risk more prominent [1]. Driving is a complex task, and different

**Data Availability Statement:** All relevant data are within the manuscript and its Supporting information files.

**Funding:** 1.This work was supported by the Science and Technology Research Program of

Chongqing Municipal Education Commission (Grant No. KJQN201900722). 2.This work was supported by the Science and Technology Bureau Foundation and Frontier Project of Chongqing (Grant No. cstc2019jcyj-msxmX0695). 3.This work was supported by National Natural Science Foundation of China(Grant No.52172341).

**Competing interests:** The authors have declared that no competing interests exist.

needs compete for a driver's attention. Suitable design of a tunnel group is favorable to driver's cognition of key information and relieves visual load. This study compares differences in the driver's gaze behavior between a single tunnel and a tunnel group and defines the high-risk factors affecting the operation of a tunnel group so as to ensure driving safety.

Tunnel groups are very common in mountain expressways. A tunnel group can constitute about 80% of an expressway, whereas the roadbed section can constitute only about 20% of an expressway. There are large differences between a tunnel group and a single tunnel in the number of tunnels and the length of distance between two tunnels, which causes a driver to experience the alternation of black and white hole effects in a short time. The cumulative visual superposition effect is obvious, which causes an excessive visual load to drivers and can easily cause traffic accidents due to uncontrolled operation. However, the current Chinese standards [2–5] and the Unified Traffic Control Equipment Manual (MUTCD) 2009 guidelines do not define the linear design specifications for tunnel groups. The existing tunnel groups are designed according to the specification requirements of a single tunnel. Therefore, how to effectively design a mountain expressway tunnel group and ensure traffic safety has become a challenge. The driver's gaze behavior can properly express the visual information processing, so the change in the driver's gaze behavior can be used to evaluate the safety and comfort of an expressway tunnel group. In this study, the difference in driver's fixation duration between a single tunnel and a tunnel group is analyzed. The Markov theory is used to determine the fixation transfer regularity and stationary distribution characteristics, and the high-risk factors affecting the operation of a tunnel group are defined to ensure driving safety.

To analyze the safety of a single tunnel and a tunnel group in mountainous expressways from the perspective of drivers, 38 tunnels, including 8 tunnel groups and 16 single tunnels, from Shuijiang to Qianjiang section are used as a research object to perform a real vehicle test and to compare differences in the drivers' fixation durations between single tunnels and tunnel groups. The k-means method is used to cluster driver's gaze points dynamically and to analyze the distribution of gaze points in different interest areas. The Markov chain, including one- and two-step gaze transition probabilities and area gaze probabilities, is used to determine the gaze transfer regularity and stationary distribution and to compare the difference between the single tunnel and tunnel group in the driver's eye gaze characteristics. The findings of this work can provide a useful basis for tunnel group design in mountainous expressways.

## 2. Literature review

### 2.1 Driving safety in single tunnel and driver's visual characteristics

Driving through a tunnel mainly relies on visual perception and access to traffic information, so the driver's eye gaze behavior can be used to determine whether the tunnel driving environment is scientific and reasonable and whether it meets the physiological needs of drivers, and this topic has drawn widespread attention in scientific research. Verwey [6] found that the closer to the tunnel a driver is, the lower the number of human gazes and the scanning range will be. Qi [7] and Yang [8] tested the driver's eye movement data in the tunnel environment. Ye [9] conducted a driving simulation test, and the pupil diameter and the sight deviation time were analyzed. He (sb' family name) [10] found that at the entrance and exit of the tunnel, drivers experienced the process of alternating cycle mutation of light environment and rapid conversion of visual light and dark adaptation. In addition, Du [11] considered the target sign's location of a driver's visual field and established the tunnel sign's visual cognition probabilistic model based on driver's eye-movement characteristics. Shang [12] employed the Markov chain to analyze the driver's eye movement characteristics to evaluate the effect of separate guide signs in a tunnel. Bai [13] analyzed the physical and psychological effects of drivers on

traffic sign recognition. Pan [14] and Wang [15] studied the relationship between the eye movement index and horizontal alignment at the entrance of a highway tunnel. Tang [16] and Huang [17] researched the colors of pavement and anti-slipping layer on driving safety at the tunnel entrance based on the eye movement indexes.

However, a single tunnel is relatively different from a tunnel group in terms of length, geometrical design, and illumination characteristics. In particular, tunnel groups on mountainous highways are more affected by the black- and white-hole effects and have a larger impact on a driver's vision and driving characteristics than a single tunnel. Therefore, it is necessary to compare driving safety of a single tunnel and a tunnel group based on the eye behavior.

## 2.2 Driving safety in tunnel group

Tunnel groups contribute local economic and social development to the infinite vitality, but there are certain challenges in determining accident-prone tunnel groups. There are large differences between tunnel group and a single tunnel, so the characteristics of traffic accidents in them are also different. Wang [18] studied the characteristics of traffic crashes in the freeway tunnel groups using a five-zone analytic approach. Xiong [19] analyzed the accident rate of the tunnel group sections of an expressway from the aspects of road alignment and the anti-sliding ability of the road surface. He (sb' family name) [20] studied the distribution of driver's attention in a mountain expressway's tunnel group sections. Yan [21, 22] analyzed the influence of the drivers' visual changes in highway tunnel groups on traffic safety. Using advanced monitoring and rescue technologies can effectively reduce the probability of tunnel groups accidents and reduce the consequences of accidents. Li [23] presented a cooperative control technology under typical emergencies in freeway tunnel groups, which was proposed to achieve coordinated cooperation between lane indicators, video monitors, and other electromechanical facilities. Guo [24] analyzed the intelligent linkage control technology to prevent traffic accidents from becoming worse and the secondary disasters from happening. Jiang [25] designed an intelligent ventilation system, which expanded the application of intelligent tunnel ventilation control systems and could provide design references to other linkage ventilation control systems of tunnel groups.

Research on traffic safety of a tunnel group has been mainly focused on analyses of accident characteristics, lighting, driving eye movement characteristics, and emergency rescue. However, an ordinary highway tunnel and an expressway tunnel group are relatively different in the speed limit, traffic composition, and lighting. In addition, there has been no comparative analysis between a tunnel group and a single tunnel, so it is impossible to know the difference between them in a driver's visual perception.

## 2.3 Markov chain application in visual gaze analysis

There has been a pressing need for quantitative comparisons of eye movement metrics [26]. Many quantitative comparison methods have been developed for two leading eye movement metrics, the scanpaths and the Markov chain, whose next state depends only on the current state and not on the past states. It works by quantizing the complexity within a system into a set of discrete states and a transition relationship between them [27]. In the eye-tracking study [28], Markov Chain models have been applied in the development of the probabilistic algorithm for gaze trajectory prediction. Chen [29] used Markov modeling to realize the eye tracking to obtain insight into a tv customer's experience. Recently, a study on the BT Ireland Innovation Centre (BTIIC) in Belfast was conducted [30], where an eye tracker was employed to evaluate customer experience of the BT Player, which is a video-on-demand application offered by the BT in the UK as part of the IPTV service, BT TV. Chatterjee [31] assumed eye

movement parameters were random variables generated by an underlying stochastic process and modeled the fixation sequences of participants using a hidden Markov model. Mauricio [32] studied how elements impacting the driver's visual behavior could provide an insight into the degree to which a vehicle's user interface influenced the attentional focus. Reimer [33] compared visual demand during the baseline driving and the driving while using voice or manual inputs to place calls with Chevrolet MyLink, and using the hidden Markov modeling to characterize the similarity of glance sequences during the two driving types. Markov chain theory has often been used to analyze drivers' visual transfer characteristics. The Markov chain can also be used to study the driver's gaze transition characteristics in a tunnel group.

Plenty of research has been conducted on tunnel traffic safety and operation management, and many results of theoretical and practical significance in scientific research have been obtained. However, there has been no definition of tunnel groups, and there has been little research on tunnel groups of mountainous expressways. At the same time, the existing relevant design specifications of a tunnel group have been mainly based on a single tunnel's technology and experience. However, the number, length, and spacing of tunnels in tunnel groups differ from those of single tunnels. In addition, there has been a lack of in-depth research on influencing mechanisms of driving safety and visual characteristics of drivers in tunnel group sections.

To analyze differences in traffic safety between single tunnels and tunnel groups with different lengths, this paper conducts real vehicle tests on the single tunnel and tunnel group using an eye tracker and obtains gaze transitions based on the area division and Markov theory.

## 3. Methods

To reflect changes in the driver's eye movement during the driving process in a tunnel, this paper presents real vehicle tests on 38 tunnels performed by five drivers. The eye tracker, non-contact multifunctional speedometer, illuminance meter, and other test equipment were used to collect the test data of the driver's fixation point, driving speed, and illuminance value in and out of the tunnel. The k-means method was used to dynamically cluster the driver's fixation point and to analyze the distribution law of a fixation point in different interest areas. Based on the area division and Markov theory, attributes related to gaze transition, including one- and two-step gaze transition probabilities and area gaze probabilities, were obtained, and differences in the eye gaze behavior between the single tunnel and tunnel group were analyzed.

### 3.1 Single tunnel and tunnel group experiments

The Shuijiang to Qianjiang section of Chongqing-Xiangyang expressway in China is 160 km long and includes two-way four-lane highways. It passes through the mountainous area and has 38 tunnels; it belongs to the typical high tunnel ratio section of the mountainous expressway. Since there is no accurate definition of a tunnel group at present, considering the specification and existing research [10, 15, 23, 24, 34, 35], in this study, a group of two or more tunnels with a spacing of less than 500 m is considered a tunnel group. Details of different tunnel type in the tunnel groups are shown in Table 1. Accordingly, the test section is divided into 8 tunnel groups and 16 single tunnels (excluding tunnel groups). Since passing through tunnels with different lengths has a great influence on driver's gaze behavior, according to the specification requirements, 16 single tunnels were divided into four categories according to the length, including two short tunnels with a length of less than 500 m, two medium tunnels with a length in the range of 500–1000 m, five long tunnels with a length in the range of 1000–3000 m, and four extra-long tunnels with a length of more than 3000 m [3]. The speed limit in the tunnel is in the range of 60–80 km/h, which is the speed limit by specification, and the

**Table 1. Details of different tunnel type in tunnel groups.**

| Tunnel type | Tunnel name | Tunnel group name | Tunnel type and spacing |
|---|---|---|---|
| Short tunnel | "Shikanzi" tunnel | "Shiziping" tunnel—**"Shikanzi" tunnel** (1#Tunnel Group) | Medium tunnel (688 m)—Spacing (493 m)—**Short tunnel (354 m)** |
| | "Wangjiaping" tunnel | "Louping" tunnel—"Tongzilin" tunnel—**"Wangjiaping" tunnel** | Medium tunnel (830 m)—Spacing (365 m)—Medium tunnel (505 m)—Spacing (110 m)—**Short tunnel (210 m)** |
| | "Zhengyang2#" tunnel | "Zhengyang1#" tunnel—**"Zhengyang2#" tunnel** | Extra-long tunnel (3285 m)—Spacing (47 m)—**Short tunnel (308 m)** |
| Medium tunnel | "Shiziping" tunnel | **"Shiziping"** tunnel—"Shikanzi" tunnel | **Medium tunnel (688 m)**—Spacing (493 m)—Short tunnel (354 m) |
| | "Wangjiangsi" tunnel | "Hujiawan" tunnel—"Dongjiawan" tunnel—**"Wangjiangsi" tunnel** | Long tunnel (688 m)—Spacing (79 m)–Long tunnel (1065 m)—Spacing (315 m)- **Medium tunnel (698 m)** |
| | "Louping" tunnel | **"Louping" tunnel**—"Tongzilin" tunnel—"Wangjiaping" tunnel | **Medium tunnel (830 m)**—Spacing (365 m)—Medium tunnel (354 m)—Spacing (110 m)–Short tunnel (210 m) |
| | "Tongzilin" tunnel | "Louping" tunnel—**"Tongzilin" tunnel**—"Wangjiaping" tunnel | Medium tunnel (830 m)—Spacing (365 m)—**Medium tunnel (354 m)**—Spacing (110 m)—Short tunnel (210m) |
| | "Tiandeng" tunnel | **"Tiandeng" tunnel**—"Banzhulin" tunnel—"Zhengyang1#" tunnel | **Medium tunnel (649 m)**—Spacing (103 m)- Extra-long tunnel (3311 m)—Spacing (345 m)- Extra-long tunnel (3600 m) |
| Long tunnel | "Dawan" tunnel | "Baima" tunnel—"Yangjiao" tunnel—**"Dawan" tunnel** | Extra-long tunnel (3144 m)—Spacing (76 m)—Extra-long tunnel (6656 m)—Spacing (210 m)–**Long tunnel (2821 m)** |
| | "Yinpan" tunnel | "Zhongxing" tunnel—**"Yinpan" tunnel** | Extra-long tunnel (6105 m)—Spacing (27 m)–**Long tunnel (1149 m)** |
| | "Hujiawan" tunnel | **"Hujiawan" tunnel**—"Dongjiawan" tunnel—"Wangjiangsi" tunnel | **Long tunnel (1201 m)**—Spacing (79 m)–Long tunnel (1065 m)—Spacing (315 m)–Medium tunnel (698 m) |
| | "Dongjiawan" tunnel | **"Dongjiawan" tunnel**—"Wangjiangsi" tunnel | **Long tunnel (1201 m)**—Spacing (79 m)–Long tunnel (1065 m)—Spacing (315 m)–Medium tunnel (698 m) |
| | "Pengshui" tunnel | **"Pengshui" tunnel**—"Changtan" tunnel | **Long tunnel (2754 m)**—Spacing (485 m)—Extra-long tunnel (3215 m) |
| | "Shaba" tunnel | **"Shaba" tunnel**—"Shihui" tunnel—"Wuling" tunnel | **Long tunnel (2736 m)**—Spacing (299 m)—Long tunnel (2226 m)- Spacing (125 m)- Long tunnel (2414 m) |
| | "Shihui" tunnel | "Shaba" tunnel—**"Shihui" tunnel**—"Wuling" tunnel | Long tunnel (2736 m)—Spacing (299 m)–**Long tunnel (2226 m)**—Spacing (125 m)–Long tunnel (2414 m) |
| | "Dongtang" tunnel | "Tiandeng" tunnel—**"Dongtang" tunnel**—"Banzhulin" tunnel—"Zhengyang1#" tunnel | Medium tunnel (649 m)—Spacing (103 m)–**Long tunnel (1687 m)**—Spacing (160 m)—Extra-long tunnel (3311 m) Spacing (345 m)—Extra-long tunnel (3600 m) |
| Extra-long tunnel | "Baima" tunnel | **"Baima" tunnel** -"Yangjiao" tunnel -"Dawan" tunnel | **Extra-long tunnel (3144 m)**—Spacing (76 m)—Extra-long tunnel (6656 m)—Spacing (210 m)—Long tunnel (2821 m) |
| | "Yangjiao" tunnel | "Baima" tunnel -**"Yangjiao" Tunnel** -"Dawan" tunnel | Extra-long tunnel (3144 m)—Spacing (76 m) **Extra-long tunnel (6656 m)**—Spacing (210 m) Long tunnel (2821 m) |
| | "Zhongxing" tunnel | **"Zhongxing" tunnel** -"Yinpan" tunnel | **Extra-long tunnel (6105 m)**—Spacing (27 m)—Long tunnel (1149 m) |
| | "Changtan" tunnel | "Xiatang" tunnel—**"Changtan" tunnel** | Long tunnel (2754 m)—Spacing (485 m)—**Extra-long tunnel (3215 m)** |
| | "Banzhulin" tunnel | "Tiandeng" tunnel—"Dongtang" tunnel—**"Banzhulin" tunnel**—"Zhengyang1#" tunnel | Medium tunnel (649 m)—Spacing (103 m)—Long tunnel (1687 m)—Spacing (160 m)—**Extra-long tunnel (3311 m)**—Spacing (345 m)—Extra-long tunnel (3600 m) |
| | "Zhengyang1#" tunnel | "Tiandeng" tunnel—"Dongtang" tunnel—"Banzhulin" tunnel—**"Zhengyang1#" tunnel** | Medium tunnel (649 m)—Spacing (103 m)—Long tunnel (1687 m)—Spacing (160 m)—Extra-long tunnel (3311 m)—Spacing (345 m)—**Extra-long tunnel (3600 m)** |

average longitudinal slope is less than 2%. There is little difference in the operating environment in the test section.

## 3.2 Participants

Five participants aged from 25 to 56 years with different driving experience were selected as test drivers. The corrected visual acuity was above 5.0, and all of them had a C1 driving license. They all performed the driver rotation test to ensure that each driver carried out at least two sets of effective tests and did not over speed. All participants in this manuscript have given

written informed consent (as outlined in PLOS consent form) to publish these case details. and had normal or corrected-to-normal vision.

## 3.3 Experimental setup

Considering the highway driving model is given priority to small vehicles, and the collection of data using test equipment is more, need a number of experimenters and testers, the Buick GL8 automatic block commercial vehicles were selected as test vehicles as shown in Fig 1. The eye movement data acquisition equipment was SMI2.1 glasses type eye tracker of SMI company from Germany. The sampling frequency of the eye tracker was 60 Hz, and the tracking accuracy was 0.1. The tracking ranges of the camera in the horizontal and vertical directions were 60˚ and 46˚, respectively. The vehicle speed acquisition was realized by a non-contact multi-function speed meter, having the measurement range and accuracy of 0–300 km/h and 0.1 km/h, respectively, which was used to determine whether the driver was speeding. The illuminance data acquisition equipment was the LX-9621 illuminance meter produced by Lantai company; its measurement range and accuracy were 0–50000 Lux and ± (5% + 5), respectively, and it was used to judge whether the lighting in each tunnel meets the design specifications. The whole driving process was recorded by a driving recorder with a resolution of 1920 × 1080 pixels, which was used to determine the consistency between the post-test calibration time and the road section location, as well as the driving location and the surrounding environment.

## 3.4 Experiment design

In order to reduce the interference of traffic volume on the test, the real vehicle test time was from 10: 00 to 16: 00 on Tuesday to Thursday, with a small traffic volume. Since the change in the light environment outside the tunnel has a great influence on the driver, the test was carried out under good weather conditions. Before the test, the driver was informed to drive according to his driving habits, comply with traffic rules, and keep the speed in the tunnel

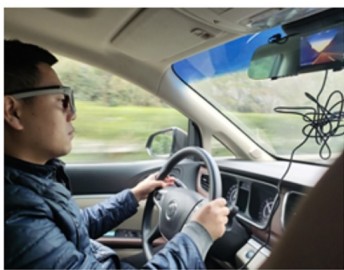
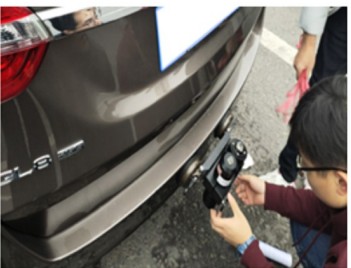
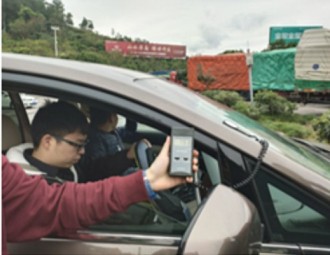
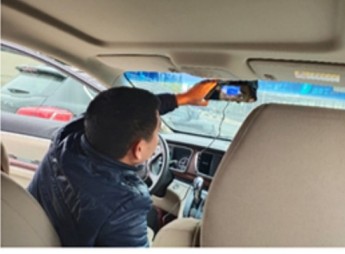

(a) eye tracker  (b) non-contact velocimetry with multi-function

(c) illuminometer  (d) driving recorder

**Fig 1. Experimental setup.**

below 80 km/h. The results of the same tunnel types in each tunnel group of 8 tunnel groups were compared to explore differences in drivers' fixation behaviors.

## 3.5 Data collection and analysis

To avoid the influence of environmental light change at the tunnel entrance on a driver, the eye movement data were extracted from the middle section of each tunnel using the BeGaze3.5 data analysis software. The basic types of the driver's eye movements included fixation, blinking, and saccade. The analysis of the driver's fixation duration during driving showed that approximately 99% of the single fixation durations were less than 1000 ms, so only the fixation samples shorter than 1000 ms were retained. The scan duration was from 30 ms to 120 ms, so the data within 100 ms were excluded. In the process of data processing, unclassified data, such as data on vehicle speeds exceeding 80 km/h, substandard lighting in tunnels, and fixation of over 1000 ms through a single gaze, were eliminated, and the driver's saccade data and fixation data were retained. Finally, the Markov model was applied to the processed data to explore the law of gaze transfer of drivers and to solve the driver's fixation point in each partition. The experimental data were screened by the Pauta criterion. From the perspective of the driver's fixation transfer characteristics, the position of the fixation point at the current moment was related to only that at the previous moment. Therefore, the Markov chain was used to analyze all the schemes [36].

# 4. Results

## 4.1 Gaze time analysis of single tunnel

Eighteen single tunnels were divided into four types: short tunnels, medium tunnels, long tunnels, and extra-long tunnels according to their lengths. The gaze time of drivers in each tunnel type was averaged. The gaze time of drivers in single tunnels of different lengths was obtained. The gaze time was classified and sampled at every 100 ms; the result is shown in Fig 2. In the

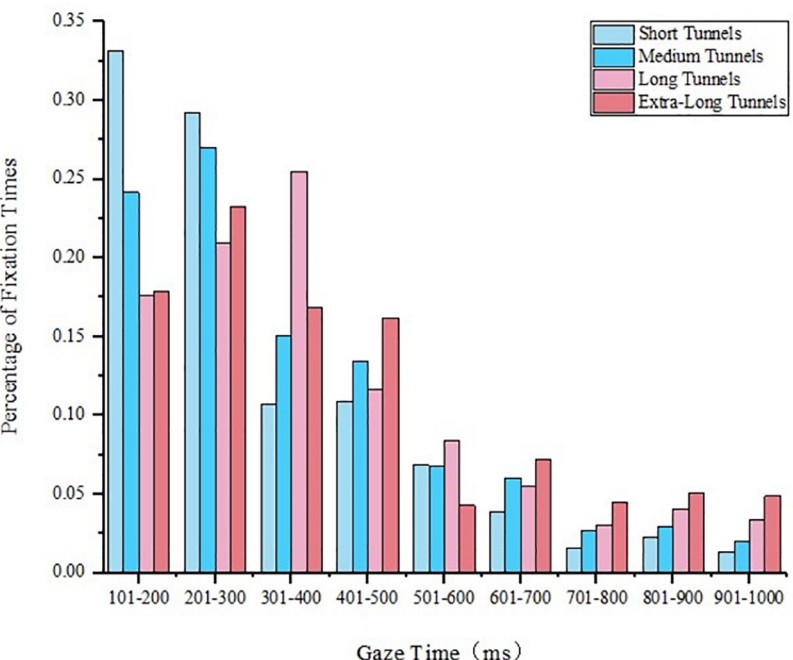

**Fig 2. Single-tunnel gaze time zoning statistical chart.**

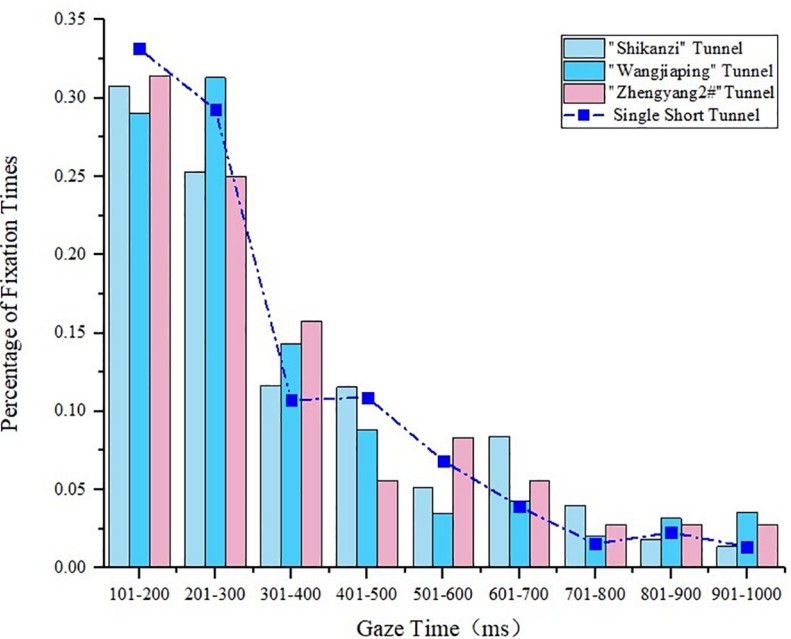

**Fig 3. Statistical diagram of the gaze time of the short tunnels in the tunnel groups.**

single tunnel, the driver's gaze time was unevenly distributed in the range of 100–1000 ms, and approximately 80% of the single gaze time was less than 500 ms; the gaze time of a single short tunnel was in the range of 0–200 ms. With the increase in the single-tunnel gaze time, the fixation frequency gradually decreased. The gaze time of a single tunnel was compared with the gaze time of a tunnel with the same length in a tunnel group.

**4.1.1 Gaze time analysis of short tunnels.** The gaze times of a single tunnel and a tunnel in a tunnel group of the same length type were compared. The gaze time of a single tunnel is shown in Fig 2. The characteristic parameters of the short tunnels in the tunnel groups are given in Table 1. The comparison of the gaze time between the single short tunnel and short tunnel in the tunnel group is presented in Fig 3, where it can be seen that, due to the short tunnel length, the tunnel in the tunnel group had a smaller number of drivers' gazes but similar gaze time compared to the single short tunnel. The gaze time of the tunnel in the tunnel group was concentrated between 200 ms and 300 ms.

**4.1.2 Gaze time analysis of medium tunnels.** The gaze time of the single medium tunnel was obtained from the results presented in Fig 2. The characteristic parameters of the medium tunnels in the tunnel groups are given in Table 1. The comparison of gaze time between the single tunnel and the tunnel in the tunnel group of medium length is presented in Fig 4. Compared with the short tunnel, the driver's gaze time of the medium tunnel was concentrated below 300 ms, and the changing trend of the gaze time of the tunnel in the tunnel group was similar to that of the single tunnel. In the tunnel group, the driver's gaze time of the medium tunnel was in the range of 700–1000 ms, which was slightly longer than that of the single medium tunnel. When the tunnel was the first tunnel in the tunnel group, its fixation time was roughly the same as that of the single tunnel.

**4.1.3 Gaze time analysis of long tunnels.** The characteristic parameters of the long tunnels in the tunnel groups are presented in Table 1. The comparison of the gaze time between the single long tunnel and the long tunnel in the tunnel group is presented in Fig 5. Compared with the short and medium tunnels, the gaze time of the long tunnels in the tunnel groups was

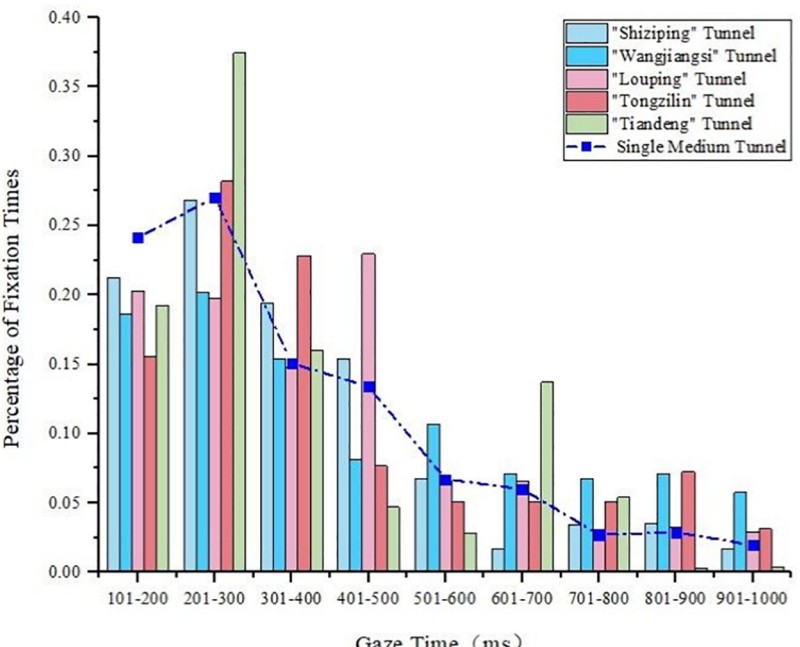

**Fig 4. Statistical diagram of the gaze time of the medium tunnels in the tunnel groups.**

significantly higher than that of the single tunnel, and it was in the range of 700–1000 ms. "Dongtang" and "Dawan" tunnels were the second tunnels in the tunnel groups, and the distance between them was close to that of the upstream tunnel. However, due to the different length of the upstream tunnels, the driver's gaze times of the two tunnels were different.

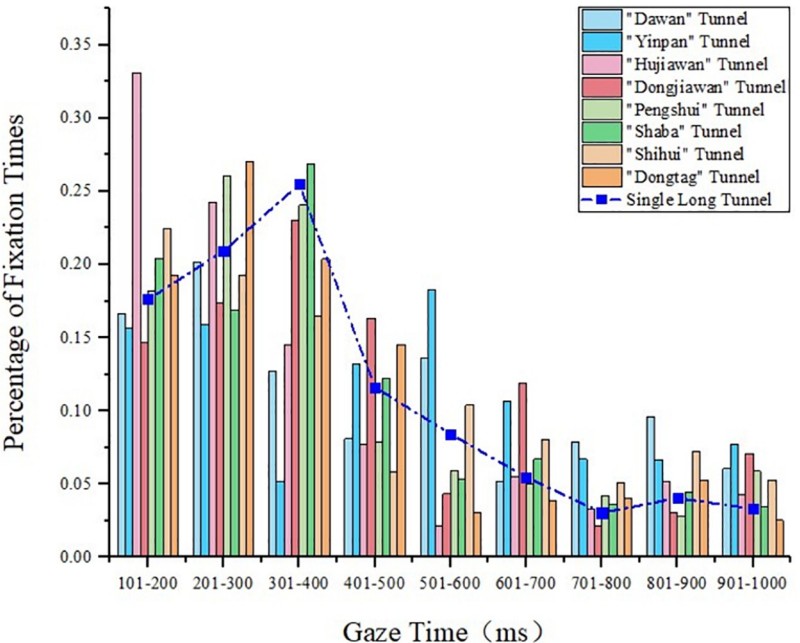

**Fig 5. Statistical diagram of the gaze time of the long tunnels in the tunnel groups.**

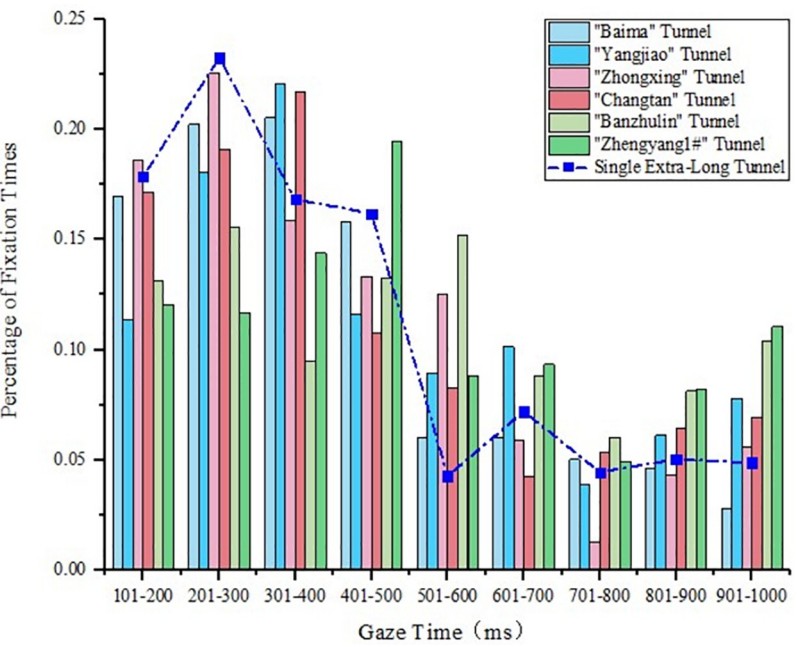

**Fig 6. Statistical diagram of the gaze time of the extra-long tunnels in the tunnel groups.**

"Dongjiawan" and "Shihui" tunnels were the second tunnels in the tunnel groups, and the upstream tunnels were long tunnels. However, due to the different distances from the upstream tunnel, the driver's gaze time was longer in the "Dongjiawan" tunnel.

**4.1.4 Gaze time analysis of extra-long tunnels.** The characteristic parameters of the extra-long tunnels in the tunnel groups are presented in Table 1. The comparison of the gaze time between the single extra-long tunnel and extra-long tunnel in the tunnel group is presented in Fig 6. The gaze time of the extra-long tunnel in the tunnel group was significantly longer than that of the single tunnel, and it was in the range of 600–1000 ms. "Banzhulin" and "Zhengyang1#" tunnels were the second and third tunnels in the same tunnel group, respectively, and since "Zhengyang1#" was more backward, their gaze times were in the range 800–1000 ms, which was significantly longer than those of the shorter tunnels. When the medium tunnel was the first tunnel in the tunnel group, the gaze time is roughly the same as that of the single tunnel, which is similar to that of the medium tunnel.

## 4.2 Gaze area division

To study the characteristics of driver's gaze point transfer, it was necessary to divide the driver's visual interest area first. In this study, the k-means dynamic clustering algorithm was used and combined with the driver's gaze target and driving characteristics to divide the driver's gaze interest area into five subareas, as shown in Fig 7. Area 1 was the cab area; Area 2 represented the near road section; Area 3 denoted the far ahead road section; Area 4 denoted the vehicle rearview mirrors, including left, right, and in-vehicle rearview mirrors; Area 5 represented all other areas, including tunnel vault and sidewall.

Aiming to pursue both left and right targets simultaneously during driving, a driver had a large deflection in the head, but the data point derived by the eye-tracker denoted a single coordinate point in a coordinate system. Therefore, to simplify the calculation, the driver's perspective was transferred into the same coordinate system.

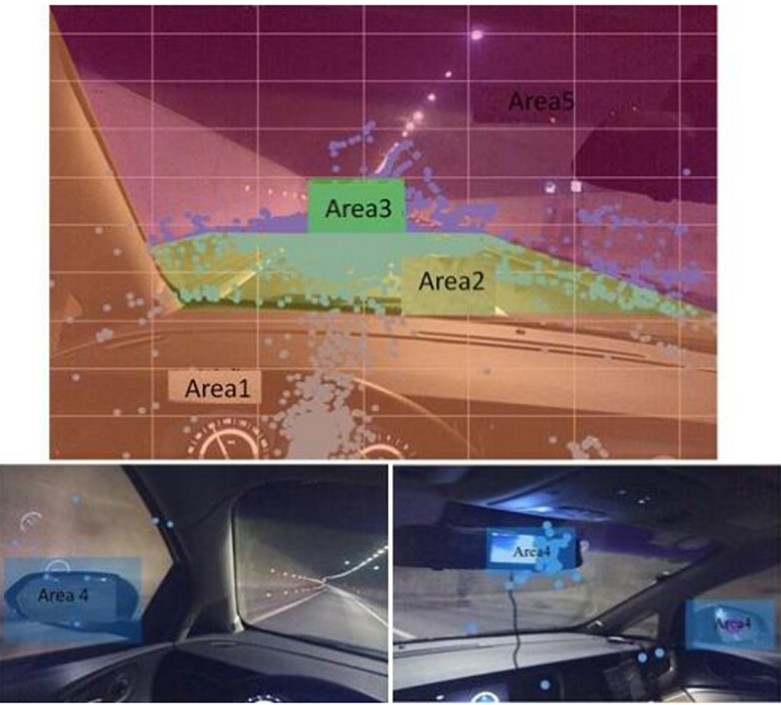

**Fig 7. Diagram of fixation area division from the driver's perspective.**

## 4.3 Gaze transfer characteristics analysis

Markov chain model is a random variable sequence, which can describe the state of a certain system, and the system state at a certain moment depends only on its state at the previous moment. In the driver's gaze behavior, the position of the driver's gaze point at a certain moment is related only to its position at the previous moment but not to the position of the other driver's gaze points. The driver's gaze transfer characteristics conform to the basic properties of the Markov chain. The driver's gaze Markov chain is discrete in terms of time and state. Therefore, the Markov chain is used in this work to analyze all schemes [36].

**4.3.1 One-step transition probability matrix of gaze behaviour.** According to the data of the eye-tracker and the division of the driver's gaze interest area, the visual attention transfer of a driver in the single tunnel was counted. After the data were processed, the Markov chain theory was used to obtain the one-step transition probability matrix of the driver's visual fixation point transformation between different regions. The one-step transition probability matrices of the single tunnels and the tunnels in the tunnel groups are presented in Table 2. The same as in the previous analysis, tunnels were divided into four groups according to their length: short, medium, long, and extra-long.

As shown in Table 2, when the tunnel was a short or medium tunnel, the probability of repeated fixation of the driver in the same area had little difference between the single tunnels and tunnels in the tunnel groups. For long tunnels, drivers had a higher probability of repeated fixation of the same area for the tunnels in the tunnel group, and the gap was significant. The number of transfer paths was defined as a number of paths with a non-zero transition probability. The number of fixation transfer paths in the long and extra-long tunnels was 17 and in the short and medium tunnels was 23. In the long tunnels, there were 24 and 23 gaze transfer paths for single tunnels and tunnels in the groups, respectively. In the extra-long tunnels, there

**Table 2. One-step transition probability matrix of different type tunnels.**

| Tunnel type | Area | Tunnel type | Area 1 | Area 2 | Area 3 | Area 4 | Area 5 |
|---|---|---|---|---|---|---|---|
| Short tunnel | Area 1 | Single short tunnel | 0.8945 | 0.0648 | 0.0407 | 0.0000 | 0.0000 |
| | | Short tunnel in tunnel group | 0.8712 | 0.0833 | 0.0455 | 0.0000 | 0.0000 |
| | Area 2 | Single short tunnel | 0.0190 | 0.9360 | 0.0415 | 0.0036 | 0.0000 |
| | | Short tunnel in tunnel group | 0.0095 | 0.9048 | 0.0857 | 0.0000 | 0.0000 |
| | Area 3 | Single short tunnel | 0.0102 | 0.0928 | 0.8714 | 0.0000 | 0.0256 |
| | | Short tunnel in tunnel group | 0.0041 | 0.0226 | 0.9680 | 0.0021 | 0.0032 |
| | Area 4 | Single short tunnel | 0.0000 | 0.2500 | 0.0000 | 0.7500 | 0.0000 |
| | | Short tunnel in tunnel group | 0.0000 | 0.0000 | 0.0000 | 0.8571 | 0.1429 |
| | Area 5 | Single short tunnel | 0.1245 | 0.0005 | 0.0023 | 0.0000 | 0.8727 |
| | | Short tunnel in tunnel group | 0.0000 | 0.1000 | 0.0417 | 0.0417 | 0.8166 |
| Medium tunnel | Area 1 | Single medium tunnel | 0.9047 | 0.0277 | 0.0625 | 0.0010 | 0.0041 |
| | | Medium tunnel in tunnel group | 0.8891 | 0.0453 | 0.0595 | 0.0045 | 0.0015 |
| | Area 2 | Single medium tunnel | 0.0350 | 0.8950 | 0.0675 | 0.0002 | 0.0024 |
| | | Medium tunnel in tunnel group | 0.0128 | 0.8312 | 0.1528 | 0.0007 | 0.0025 |
| | Area 3 | Single medium tunnel | 0.0178 | 0.0476 | 0.9281 | 0.0013 | 0.0051 |
| | | Medium tunnel in tunnel group | 0.0215 | 0.0347 | 0.9343 | 0.0014 | 0.0080 |
| | Area 4 | Single medium tunnel | 0.0008 | 0.0270 | 0.0000 | 0.6597 | 0.3125 |
| | | Medium tunnel in tunnel group | 0.1083 | 0.0639 | 0.0000 | 0.7444 | 0.0833 |
| | Area 5 | Single medium tunnel | 0.0000 | 0.0336 | 0.0336 | 0.0263 | 0.9066 |
| | | Medium tunnel in tunnel group | 0.0082 | 0.1227 | 0.0742 | 0.0000 | 0.7949 |
| Long tunnel | Area 1 | Single long tunnel | 0.8926 | 0.0401 | 0.0645 | 0.0004 | 0.0024 |
| | | Long tunnel in tunnel group | 0.8910 | 0.0202 | 0.0841 | 0.0026 | 0.0021 |
| | Area 2 | Single long tunnel | 0.0175 | 0.9426 | 0.0328 | 0.0036 | 0.0034 |
| | | Long tunnel in tunnel group | 0.0331 | 0.9090 | 0.0283 | 0.0086 | 0.0210 |
| | Area 3 | Single long tunnel | 0.0125 | 0.0685 | 0.9046 | 0.0040 | 0.0103 |
| | | Long tunnel in tunnel group | 0.0176 | 0.0200 | 0.9548 | 0.0021 | 0.0055 |
| | Area 4 | Single long tunnel | 0.0083 | 0.0886 | 0.0083 | 0.8528 | 0.0420 |
| | | Long tunnel in tunnel group | 0.0000 | 0.0695 | 0.0205 | 0.8372 | 0.0728 |
| | Area 5 | Single long tunnel | 0.0357 | 0.0706 | 0.1750 | 0.0000 | 0.7187 |
| | | Long tunnel in tunnel group | 0.0188 | 0.0353 | 0.0727 | 0.0000 | 0.8732 |
| Extra-long tunnel | Area 1 | Single extra-long tunnel | 0.8860 | 0.0655 | 0.0400 | 0.0028 | 0.0057 |
| | | Extra-long tunnel in tunnel group | 0.8929 | 0.0603 | 0.0442 | 0.0014 | 0.0012 |
| | Area 2 | Single extra-long tunnel | 0.0280 | 0.9474 | 0.0183 | 0.0024 | 0.0039 |
| | | Extra-long tunnel in tunnel group | 0.0144 | 0.9389 | 0.0395 | 0.0030 | 0.0041 |
| | Area 3 | Single extra-long tunnel | 0.0151 | 0.0619 | 0.9122 | 0.0024 | 0.0083 |
| | | Extra-long tunnel in tunnel group | 0.0132 | 0.0621 | 0.9143 | 0.0009 | 0.0094 |
| | Area 4 | Single extra-long tunnel | 0.0217 | 0.0545 | 0.0961 | 0.7005 | 0.1272 |
| | | Extra-long tunnel in tunnel group | 0.0000 | 0.0778 | 0.0069 | 0.7996 | 0.1157 |
| | Area 5 | Single extra-long tunnel | 0.0067 | 0.1440 | 0.0463 | 0.0337 | 0.7693 |
| | | Extra-long tunnel in tunnel group | 0.0196 | 0.1297 | 0.0261 | 0.0102 | 0.8144 |

was no path with a transition probability of zero for single tunnels, and there was only one path with a transition probability of zero for the tunnel groups.

**4.3.2 Two-step transition probability matrix of gaze behaviour.** According to the one-step transition probability matrix of a driver, the two-step transition probability matrix of the driver's gaze point was obtained. The two-step transition calculation was performed for the

**Table 3. Two-step transition probability matrix of different type tunnels.**

| Tunnel type | Area | Tunnel type | Area 1 | Area 2 | Area 3 | Area 4 | Area 5 |
|---|---|---|---|---|---|---|---|
| Short tunnel | Area 1 | Single short tunnel | 0.8030 | 0.1188 | 0.0780 | 0.0002 | 0.0000 |
| | | Short tunnel in tunnel group | 0.7616 | 0.1448 | 0.0933 | 0.0002 | 0.0000 |
| | Area 2 | Single short tunnel | 0.0343 | 0.8074 | 0.0698 | 0.0861 | 0.0024 |
| | | Short tunnel in tunnel group | 0.0166 | 0.8241 | 0.1588 | 0.0000 | 0.0006 |
| | Area 3 | Single short tunnel | 0.0246 | 0.1535 | 0.7816 | 0.0001 | 0.0402 |
| | | Short tunnel in tunnel group | 0.0081 | 0.0451 | 0.9397 | 0.0042 | 0.0031 |
| | Area 4 | Single short tunnel | 0.0018 | 0.4286 | 0.0054 | 0.5642 | 0.0000 |
| | | Short tunnel in tunnel group | 0.0000 | 0.0000 | 0.0119 | 0.7511 | 0.2370 |
| | Area 5 | Single short tunnel | 0.2205 | 0.0139 | 0.0000 | 0.0000 | 0.7656 |
| | | Short tunnel in tunnel group | 0.0051 | 0.1824 | 0.1187 | 0.0346 | 0.6592 |
| Medium tunnel | Area | Tunnel type | Area 1 | Area 2 | Area 3 | Area 4 | Area 5 |
| | Area 1 | Single medium tunnel | 0.8221 | 0.0537 | 0.1161 | 0.0001 | 0.0080 |
| | | Medium tunnel in tunnel group | 0.7949 | 0.0820 | 0.1134 | 0.0067 | 0.0030 |
| | Area 2 | Single medium tunnel | 0.0631 | 0.8072 | 0.1244 | 0.0003 | 0.0050 |
| | | Medium tunnel in tunnel group | 0.0262 | 0.7021 | 0.2648 | 0.0013 | 0.0056 |
| | Area 3 | Single medium tunnel | 0.0337 | 0.0882 | 0.8657 | 0.0021 | 0.0103 |
| | | Medium tunnel in tunnel group | 0.0401 | 0.0638 | 0.8793 | 0.0027 | 0.0141 |
| | Area 4 | Single medium tunnel | 0.0003 | 0.0629 | 0.0128 | 0.4664 | 0.4575 |
| | | Medium tunnel in tunnel group | 0.1579 | 0.1073 | 0.0226 | 0.5816 | 0.1307 |
| | Area 5 | Single medium tunnel | 0.0019 | 0.0631 | 0.0644 | 0.0452 | 0.8255 |
| | | Medium tunnel in tunnel group | 0.0194 | 0.0969 | 0.0300 | 0.0003 | 0.8534 |
| Long tunnel | Area 1 | Single long tunnel | 0.7990 | 0.0763 | 0.1191 | 0.0010 | 0.0046 |
| | | Long tunnel in tunnel group | 0.7962 | 0.0380 | 0.1564 | 0.0045 | 0.0049 |
| | Area 2 | Single long tunnel | 0.0327 | 0.8919 | 0.0627 | 0.0063 | 0.0064 |
| | | Long tunnel in tunnel group | 0.0578 | 0.8456 | 0.0447 | 0.0147 | 0.0372 |
| | Area 3 | Single long tunnel | 0.0243 | 0.1252 | 0.8262 | 0.0072 | 0.0171 |
| | | Long tunnel in tunnel group | 0.0331 | 0.0375 | 0.9150 | 0.0038 | 0.0106 |
| | Area 4 | Single long tunnel | 0.0180 | 0.1598 | 0.0259 | 0.7290 | 0.0673 |
| | | Long tunnel in tunnel group | 0.0030 | 0.1157 | 0.0555 | 0.7987 | 0.0271 |
| | Area 5 | Single long tunnel | 0.0524 | 0.1264 | 0.2731 | 0.0009 | 0.5472 |
| | | Long tunnel in tunnel group | 0.0174 | 0.0638 | 0.1340 | 0.0163 | 0.7685 |
| | Area | Tunnel Type | Area 1 | Area 2 | Area 3 | Area 4 | Area 5 |
| | Area 1 | Single extra-long tunnel | 0.7873 | 0.1229 | 0.0736 | 0.0052 | 0.0110 |
| | | Extra-long tunnel in tunnel group | 0.7993 | 0.1125 | 0.0833 | 0.0025 | 0.0024 |
| Extra-long tunnel | Area 2 | Single extra-long tunnel | 0.0510 | 0.9012 | 0.0356 | 0.0046 | 0.0076 |
| | | Extra-long tunnel in tunnel group | 0.0270 | 0.8854 | 0.0751 | 0.0054 | 0.0071 |
| | Area 3 | Single extra-long tunnel | 0.0286 | 0.1162 | 0.8351 | 0.0045 | 0.0156 |
| | | Extra-long tunnel in tunnel group | 0.0248 | 0.0160 | 0.9420 | 0.0021 | 0.0151 |
| | Area 4 | Single extra-long tunnel | 0.0386 | 0.1103 | 0.0156 | 0.6726 | 0.1629 |
| | | Extra-long tunnel in tunnel group | 0.0031 | 0.1790 | 0.0241 | 0.7028 | 0.0909 |
| | Area 5 | Single extra-long tunnel | 0.0152 | 0.2380 | 0.0828 | 0.0465 | 0.6176 |
| | | Extra-long tunnel in tunnel group | 0.0349 | 0.1177 | 0.1162 | 0.0158 | 0.7154 |

five areas, and the two-step transition probability matrices of the four tunnel types were obtained, as shown in Table 3.

The numbers of driver's gaze transfer paths in the single short tunnels and short tunnels in the groups were 21. In tunnels of the other lengths, there was no path with a transition

probability of zero. If the initial position of a fixation point falls in the same region, that is called "looking back" phenomenon after two transition steps. The two-step transition probability of the driver's gaze in this region is the maximum, and it is higher than those of the other regions. The probability of the driver looking back at the same area in the tunnels in the tunnel groups was slightly higher than that in the single tunnels. With the increase in the tunnel length, the probability of looking back in areas 2 and 3 increased, and in the extra-long tunnels in the tunnel groups, this probability reached 0.9420.

**4.3.3 Stable distribution of gaze behaviour.** The stationary distribution probabilities of different tunnel types were calculated.

$$\pi_{Single\ short\ tunnel} = \begin{bmatrix} 0.1648 \\ 0.4048 \\ 0.4007 \\ 0.0061 \\ 0.0236 \end{bmatrix} \pi_{Single\ medium\ tunnel} = \begin{bmatrix} 0.1542 \\ 0.3320 \\ 0.4388 \\ 0.0102 \\ 0.0648 \end{bmatrix} \pi_{Single\ long\ tunnel} = \begin{bmatrix} 0.1403 \\ 0.4085 \\ 0.4030 \\ 0.0170 \\ 0.0312 \end{bmatrix} \pi_{Single\ extra-long\ tunnel} = \begin{bmatrix} 0.1317 \\ 0.4883 \\ 0.2953 \\ 0.0205 \\ 0.0642 \end{bmatrix}$$

$$\pi_{Short\ tunnel\ in\ tunnel\ groups} = \begin{bmatrix} 0.0578 \\ 0.1847 \\ 0.6891 \\ 0.0399 \\ 0.0286 \end{bmatrix} \pi_{Medium\ tunnel\ in\ tunnel\ groups} = \begin{bmatrix} 0.1788 \\ 0.2127 \\ 0.5430 \\ 0.0112 \\ 0.0543 \end{bmatrix}$$

$$\pi_{Long\ tunnel\ in\ tunnel\ groups} = \begin{bmatrix} 0.1427 \\ 0.1786 \\ 0.6034 \\ 0.0191 \\ 0.0562 \end{bmatrix} \pi_{Extra-long\ tunnel\ in\ tunnel\ groups} = \begin{bmatrix} 0.1237 \\ 0.3904 \\ 0.4415 \\ 0.0125 \\ 0.0319 \end{bmatrix}$$

Regardless of the tunnel type (single tunnel or tunnel in a group), drivers paid the most attention to areas 2 and 3. With the increase in the tunnel length, the driver's attention to these two areas increased, and it was higher for the tunnels in the groups than for the single tunnels.

## 5. Discussion

Through the analysis of domestic and foreign literature, it is found that domestic and foreign scholars mainly select visual indicators such as fixation time, fixation point density and pupil area to analyze the visual load of drivers in single tunnels and tunnel groups respectively. This study analyzes the difference between mountainous-expressway single tunnels and tunnel groups based on eye gaze behavior. Compared with the single tunnel, the fixation time of drivers in the tunnel groups with the same length was longer, and it was in the range of 700–1000 ms. This indicated that the driver's visual load increased, and it took longer to identify objects in the tunnel. With the increase in the tunnel length, the frequency of the driver's concentrated gaze showed an upward trend, from 200 ms to 400 ms, which also indicated that the driver's psychological pressure in the tunnel group increased. When the tunnel length was the same, the driver's position in the tunnel group was more backward, and the fixation duration was longer. The reason was that the driver had experienced multiple tunnels, so the cumulative

visual effect increased. However, this was related to the tunnel position in the tunnel group. If the interval between tunnels was short, the driver could not adapt to the external environment of the tunnel. The frequent "black hole" and "white hole" effects led to a longer time needed to identify objects in the downstream tunnel. In addition, the length of the upstream tunnel also affected the driving safety of the downstream tunnel in the tunnel group. The longer the upstream tunnel length, the longer the driver's gaze duration in the downstream tunnel.

As shown in Table 2, in long and extra-long tunnels, the probability of driver's repeated fixation in the same area was significantly different for single tunnels and tunnels in group. The results showed that it was more difficult for drivers to obtain information in long and extra-long tunnels than in medium and short tunnels, and the probability of repeated gaze on the target was higher. In the short tunnels, the gaze transfer paths were short because the tunnel length was short, and the driving time through the tunnel was also short, so the scope of driver's attention was limited. Two-step gaze transition probability can reflect a continuous transition behavior. As shown in Table 3, the driver gazed at the short transition path in the short tunnel because this behavior was the reflection of the one-step transition probability was the same as the one-step transition probability. The driver's two-step transition probability of looking back to the same region was the largest, which was related to the specific environment of a single tunnel. The two-step transition probability of drivers in tunnel groups was larger than that in single tunnels, indicating that the drivers needed to confirm the region repeatedly. With the increase in the tunnel length, the two-step transition probabilities of drivers in areas 2 and 3 showed an upward trend, and this increase in the tunnel groups was more significant. The two-step transition probability in the extra-long tunnels in the tunnel groups was 0.9420, which can be explained as follows. Area 3 included the tunnel entrance, which is far ahead, indicating that the driver had been looking for the tunnel exit and was anxious to leave the tunnel. In the stationary distribution of the gaze behaviour, with the increase in the tunnel length, the driver's attention to areas 2 and 3 increased, and it was higher for the tunnel group than for the single tunnel. However, the driver paid too much attention to these two areas, which led to the neglect of the other areas. However, the lack of attention to traffic speed and surrounding traffic conditions is prone to visual fatigue and safety hazards, which can cause traffic accidents.

In general, when the tunnel length was shorter than 1000 m, which included short and medium tunnels, there were little differences in the fixation time and fixation point transfer between the single tunnels and the tunnel groups; thus, in this case, the safety of the tunnel groups does not require special consideration. When the tunnel length was longer than 1000 m, which included long and extra-long tunnels, the repeated fixation probability and two-step transition probability of drivers in the tunnel groups were higher than those in the single tunnels, which indicated that the special environment of the tunnel group had a great influence on the driver's fixation characteristics. Thus, for these tunnels, it is necessary to pay special attention to driving safety, and the design and management methods of single tunnels cannot be applied directly. Particularly, for extra-long tunnels, which were located in the back of the tunnel groups, the length of the upstream tunnel was long, and the distance between the upstream tunnel and the tunnel was short; thus, the driver was passing through multiple tunnels continuously, driving in tunnels for a long time, and frequently experiencing the effect of black and white holes, so the cumulative visual pressure could not be effectively released. Considering that the superposition effect of psychology and vision can easily increase the driving risk, the design and management of these tunnels require further research and analysis. The results of this paper are similar to [37, 38], they all show that there is a significant difference between the driver' s gaze characteristics of the single tunnel and tunnel group.

## 6. Conclusion

To analyze the difference between tunnel groups and single tunnels in terms of operational safety, this paper conducts a real vehicle test. The test results show that when the tunnel length is short or medium, there are little differences in the fixation time and fixation point transfer between the two tunnel types, and the safety of the tunnel groups does not require special consideration. When the tunnel length is longer than 1000 m, the repeated fixation probability and two-step transition probability of drivers in the tunnel groups are higher than those in the single tunnels; thus, for these tunnels, it is necessary to consider driving safety further. Particularly, for extra-long tunnels located at the back of the tunnel groups, the design and management of these tunnels require special attention. The methods and techniques for improving the driving safety of single tunnels have been widely studied, but there have been fewer studies on mountain expressway tunnels. Therefore, this study can provide a helpful reference for understanding the difference in driving safety between single tunnels and tunnels in a group. The presented research results can also provide a theoretical basis for the design and management of long and extra-long tunnels in the mountainous expressway. The main limitation of this study is that it does not consider the influence of tunnel alignment on a driver's visual characteristics. The curve section in tunnel groups could be considered in future work. In addition, there are only eight tunnel groups in the real vehicle test section in this paper. Due to the limited conditions, it is impossible to test each possible combination of tunnel groups, and the research on the combination form of tunnel groups can be added in further research.

## Supporting information

**S1 Table. Transition probability matrix of gaze behavior.**
(PDF)

## Acknowledgments

We thank the associate editor and the reviewers for their useful feedback that improve this paper. In addition, we thank the research group for their help in experimental equipment and technology.

## Author Contributions

**Conceptualization:** Ting Shang, Hongjiao Qi, An Huang, Tangzhi Liu.

**Data curation:** Hongjiao Qi, An Huang.

**Formal analysis:** Ting Shang, Tangzhi Liu.

**Writing – original draft:** Hongjiao Qi.

**Writing – review & editing:** Ting Shang, Hongjiao Qi, An Huang, Tangzhi Liu.

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
