## [Decision Letter · Decision Letter 0]

23 Dec 2021

PONE-D-21-35527A Comparative Driving Safety Study of Mountainous Expressway Individual Tunnel and Tunnel Group Based on Eye Gaze BehaviorPLOS ONE

Dear Dr. Huang,

Thank you for submitting your manuscript to PLOS ONE. After careful consideration, we feel that it has merit but does not fully meet PLOS ONE’s publication criteria as it currently stands. Therefore, we invite you to submit a revised version of the manuscript that addresses the points raised during the review process.

We look forward to receiving your revised manuscript.

Kind regards,

Feng Chen

Academic Editor

PLOS ONE

Journal Requirements:

( Yes. 1.This work was supported by the Science and Technology Research Program of Chongqing Municipal Education Commission (Grant No. KJQN201900722) and the Science and Technology Bureau Foundation and Frontier Project of Chongqing (Grant No. cstc2019jcyj-msxmX0695). We thank LetPub (www.letpub.com) for its linguistic assistance during the preparation of this manuscript.

2.This work was supported by the Science and Technology Research Program of Chongqing Municipal Education Commission (Grant No. KJQN201900722) and the Science and Technology Bureau Foundation and Frontier Project of Chongqing (Grant No. cstc2019jcyj-msxmX0695). 

3.National Natural Science Foundation of China(Grant No.52172341))

(NO authors have competing interests.)

5. Please upload a copy of Figure 8, to which you refer in your text on page 13. If the figure is no longer to be included as part of the submission please remove all reference to it within the text.

6. We note that Figure 1 in your submission contain satellite images which may be copyrighted. All PLOS content is published under the Creative Commons Attribution License (CC BY 4.0), which means that the manuscript, images, and Supporting Information files will be freely available online, and any third party is permitted to access, download, copy, distribute, and use these materials in any way, even commercially, with proper attribution. For these reasons, we cannot publish previously copyrighted maps or satellite images created using proprietary data, such as Google software (Google Maps, Street View, and Earth). For more information, see our copyright guidelines: http://journals.plos.org/plosone/s/licenses-and-copyright.

7. Thank you for stating the following in the Acknowledgments Section of your manuscript: 

( This work was supported by the Science and Technology Bureau Foundation and Frontier Project of Chongqing (cstc2019jcyj-msxmX0695), Science and Technology Research Program of Chongqing Municipal Education Commission (Grant No. KJQN201900722).)

(Yes. 

a.This work was supported by the Science and Technology Research Program of Chongqing Municipal Education Commission (Grant No. KJQN201900722) and the Science and Technology Bureau Foundation and Frontier Project of Chongqing (Grant No. cstc2019jcyj-msxmX0695). We thank LetPub (www.letpub.com) for its linguistic assistance during the preparation of this manuscript.

b.This work was supported by the Science and Technology Research Program of Chongqing Municipal Education Commission (Grant No. KJQN201900722) and the Science and Technology Bureau Foundation and Frontier Project of Chongqing (Grant No. cstc2019jcyj-msxmX0695). 

c.National Natural Science Foundation of China(Grant No.52172341))

8. We note that Figure 2 includes an image of a participant

Reviewers' comments:

Reviewer's Responses to Questions

**Comments to the Author**

1. Is the manuscript technically sound, and do the data support the conclusions?

Reviewer #1: Yes

Reviewer #2: Yes

2. Has the statistical analysis been performed appropriately and rigorously? 

Reviewer #1: Yes

Reviewer #2: Yes

3. Have the authors made all data underlying the findings in their manuscript fully available?

Reviewer #1: Yes

Reviewer #2: Yes

4. Is the manuscript presented in an intelligible fashion and written in standard English?

Reviewer #1: Yes

Reviewer #2: Yes

5. Review Comments to the Author

Reviewer #1: It is an interesting article. The authors conducted an extraordinary amount of field tests to study the eye gaze characteristics along a single tunnel and tunnel group. Please take the following recommendations into consideration:

1. It is better to specify the content of “suitable design”, just a brief presentation., which could be provide an information about the relationship between eye gaze characteristic and “which part of the design”.

2. Introduction line 44-48 on page 2 “The cumulative visual superposition effect is obvious, which causes an excessive visual load to drivers and can easily cause traffic accidents due to uncontrolled operation.” Is there any study supporting this claim?

3. The author often discusses about the difference of the gaze transition probability between individual tunnel and tunnel group but it is better to test the difference statistically (statistically significant or not).

4. It is good to mention the power of the Markov model used.

5. In the discussion part, please discuss published studies that show similar or dissimilar trends should be discussed.

6. Please add possible ideas for further research in the future, to line 446-448 on page 20. Elaborate, not only mention the curve section.

Reviewer #2: Generally, this is a qualified paper focused on the drivers' eye gaze behavior in tunnels of mountainous expressways. The research was conducted by a number of on-site tests that collected a lot of solid data. The research contents are proper, the methodology is reasonable and the conclusion is valid.

The following comments are used for evisions' reference.

1, Please explain why a 500m interval between two adjacent tunneal is used as threshold for defining tunnel group?

2, The definitions for short, medium, long and extra-long tunnels can be moved from "3.4 Experiment Design, L222~225" upward like "3.1 Single Tunnel and Tunnel Group Experiments", for an easier understanding.

3, L351, The two-step transfer, or The two-step transition? which word is better? like say, in the whole paper, are "transfer" and "transition" the same? could be replaceable? or used in the fixed terms each other? Please note clearly.

4, L437 "repetition" should be revised as "repeated".

5, Some comments are marked in the original paper.

Finally, English in the paper should be further polished， in order to be accurate and concise.

6. PLOS authors have the option to publish the peer review history of their article (what does this mean?). If published, this will include your full peer review and any attached files.

Reviewer #1: No

Reviewer #2: No

---

## [Author Response · Author response to Decision Letter 0]

16 Jan 2022

Response to Reviewers

Dear Editor and Reviewers:

Thank you for your letter and reviewers’ comments concerning our manuscript entitled “A Comparative Driving Safety Study of Mountainous Expressway Individual Tunnel and Tunnel Group Based on Eye Gaze Behavior”. Those comments are all valuable and very helpful for revising and improving our paper, as well as the important guiding significance to our researches. We have studied your letter, and revised the manuscript which we hope meet with approval. Here below is our description on revision according to journal requirements and the reviewers’ comments.

According to journal requirements, we make the following statement. Firstly, we have modified the manuscript style templates to meet PLOS ONE's style requirements. Secondly, this work was supported by the Science and Technology Bureau Foundation and Frontier Project of Chongqing (cstc2019jcyj-msxmX0695), Science and Technology Research Program of Chongqing Municipal Education Commission (Grant No. KJQN201900722), National Natural Science Foundation of China (Grant No.52172341)), and there was no additional external funding received for this study, please refer to Annex 1 for the Funding Statement. Thirdly, the authors have declared that no competing interests exist. The above two points have been modified in the cover letter. The fourth, as for the use of pictures in the paper, we have removed Figure 1 from the manuscript and uploaded the copy of Figure 8(Figure 7 in the revised manuscript) see Annex 2 and. The fifth, we have uploaded the documents of consent form related to Figure 2(Figure 1 in the revised manuscript), as shown in Annex 3. In addition, we have removed any funding-related text from the manuscript and added caption for Supporting Information files at the end of the manuscript. Finally, we have reviewed our reference list to ensure that it is complete and correct. 

The main correction in the paper and the responds to the reviewer’s comment are as flowing:

Response to the reviewer’s comments: 

Reviewer #1: 

1. It is better to specify the content of “suitable design”, just a brief presentation, which could be provide an information about the relationship between eye gaze characteristic and “which part of the design”.

Response: Thank you very much for your suggestion. The analysis of driver's eye gaze characteristics, such as gaze time and gaze area division are to provide data basis for gaze probability transition in section 4.3.

2. Introduction line 44-48 on page 2 “The cumulative visual superposition effect is obvious, which causes an excessive visual load to drivers and can easily cause traffic accidents due to uncontrolled operation.” Is there any study supporting this claim?

Response: Thank you very much for your questions. Yes, there are studies to support this claim. For example, Duan Mengmeng put forward the concept of " load repeat accumulation effect" to express the phenomenon of visual changes of drivers in the process of frequent access to tunnels in his research on Drive’s Visual Load at Tunnel Entrance and Exit of Sections with High Ratio of Tunnels, and pointed out that this phenomenon causes an excessive visual load to drivers and is extremely detrimental to driving safety. The " load repeat accumulation effect" proposed above is the similar concept as "The cumulative visual superposition effect " mentioned in this paper.

[1] Duan, M. M, Tang, B. M., Hu X. H., etc. “Research on driver's visual load at tunnel entrance and exit of high tunnel ratio section”. Transportation system engineering and information, vol. 18, pp. 113-119, 2018.

3. The author often discusses about the difference of the gaze transition probability between individual tunnel and tunnel group but it is better to test the difference statistically (statistically significant or not).

Response: Thank you very much for your suggestion. Your suggestion that the use of statistical significance test is a good way to further validate the difference between single tunnel and tunnel group driver’s fixation transfer. From the one-step transfer matrix, two-step transfer matrix and steady-state matrix calculated by the Markov theory in this paper, it can be seen that the probability values of the single tunnel and tunnel group in the short, medium, long and extra-long four types of driver’s fixation points to different regions are distributed between 0～1, and the difference is not significant. The sample size is small, and the proportion of ‘0’ value in the one-step, two-step and steady-state matrix is high, so it is not suitable for the analysis of statistical significance test. In addition, we reviewed the relevant literature and found that scholars used Markov theory to study the driver's fixation transfer characteristics did not use statistical difference test for comparative analysis[2][3]. However, the suggestions you put forward are very meaningful, and we will continue to explore the methods that can analyze the difference of the driver’s fixation transfer distribution in single tunnels and tunnel groups in subsequent studies.

[2] Peng, J. S., Wang, C. W., Fu, R., etc. “Extraction of parameters for lane change intention based on driver's gaze transfer characteristics”. Safety Science, vol. 126 :104647. 

[3] Guo, TI, Pan Shu, Shao Fei, etc. “Characteristics of urban tunnel driver's gaze behavior”. Journal of Southeast University (English Edition), vol. 37, pp. 325-331, 2021.

4. It is good to mention the power of the Markov model used.

Response: Thank you very much for your questions. Markov model has typical characteristics of stochastic process without aftereffect. To determine the state of the model at “T+1”, we just need to know its state at “T”, and do not need to know its previous state. Similarly, the driver in the driving process, the next fixation in the region of interest is only related to the current state, and has nothing to do with the previous fixation. If the sweep behavior is removed, all the fixation is discrete in time and state. Therefore, the use of discrete Markov chain theory can solve the problem well.

5. In the discussion part, please discuss published studies that show similar or dissimilar trends should be discussed.

Response: Thank you very much for your suggestion. We have discussed the similar trends between this paper and published studies in the discussion section. Please refer to section 5 of the file named ‘Revised Manuscript with Track Changes’ for details.

6. Please add possible ideas for further research in the future, to line 446-448 on page 20. Elaborate, not only mention the curve section.

Response: Thank you very much for your suggestion. We have added possible ideas for further research in the future. We have marked the corrections in red in the manuscript. Please refer to section 6 of the file named ‘Revised Manuscript with Track Changes’ for details.

Reviewer #2: 

1. Please explain why a 500m interval between two adjacent tunnel is used as threshold for defining tunnel group?

Response：Thank you very much for your questions. According to the early research group, we define the tunnel group from the perspective of the driver's visual characteristics. Answer in detail that a file named "Respond to Reviewers " has been uploaded.

2. The definitions for short, medium, long and extra-long tunnels can be moved from "3.4 Experiment Design, L222~225" upward like "3.1 Single Tunnel and Tunnel Group Experiments", for an easier understanding.

Response: Thank you very much for your suggestion. We have correction according to the Reviewer’s suggestion. Please refer to ‘Revised Manuscript with Track Changes’ for details.

3. L351, The two-step transfer, or The two-step transition? which word is better? like say, in the whole paper, are "transfer" and "transition" the same? could be replaceable? or used in the fixed terms each other? Please note clearly.

Response: We looked through a lot of relevant references and found that “transition” is the most popular word for the “Markov transition probability”. Therefore, "The two-step transition" is used in this manuscript. In order to ensure the consistency of the words used throughout the manuscript, about “Markov transition”, we all use “transition”. In addition, “transfer” is commonly used in the “fixation point transfer; gaze transfer; visual transfer, etc”. We have marked the corrections in red in the manuscript. Please refer to ‘Revised Manuscript with Track Changes’ for details.

4. L437 "repetition" should be revised as "repeated".

Response: Thank you very much for your suggestion, we have revised the text according to the reviewer 's suggestion. Please refer to the file named ‘Revised Manuscript with Track Changes’ for details.

5. Some comments are marked in the original paper.

Response: Thank you very much for your suggestion. We modified the article according to your comments that are marked in the original paper. Please refer to ‘Revised Manuscript with Track Changes’ for details.

Special thanks to you for your good comments.

We tried our best to improve the manuscript and made some change in the manuscript. These changes will not influence the content and framework of the paper. And here we did not list the changes but marked in red in the file named ' Revised Manuscript with Track Changes '. 

We appreciate for Editor and Reviewers’ warm work earnestly, and hope that the correction will meet with approval.

Once again, thank you very much for your comments and suggestion.

---

## [Decision Letter · Decision Letter 1]

28 Jan 2022

A Comparative Driving Safety Study of Mountainous Expressway Individual Tunnel and Tunnel Group Based on Eye Gaze Behavior

PONE-D-21-35527R1

Dear Dr. Huang,

We’re pleased to inform you that your manuscript has been judged scientifically suitable for publication and will be formally accepted for publication once it meets all outstanding technical requirements.

Kind regards,

Feng Chen

Academic Editor

PLOS ONE

Additional Editor Comments (optional):

Reviewers' comments:

Reviewer's Responses to Questions

**Comments to the Author**

1. If the authors have adequately addressed your comments raised in a previous round of review and you feel that this manuscript is now acceptable for publication, you may indicate that here to bypass the “Comments to the Author” section, enter your conflict of interest statement in the “Confidential to Editor” section, and submit your "Accept" recommendation.

Reviewer #1: All comments have been addressed

Reviewer #2: All comments have been addressed

2. Is the manuscript technically sound, and do the data support the conclusions?

Reviewer #1: Yes

Reviewer #2: Yes

3. Has the statistical analysis been performed appropriately and rigorously? 

Reviewer #1: Yes

Reviewer #2: Yes

4. Have the authors made all data underlying the findings in their manuscript fully available?

Reviewer #1: Yes

Reviewer #2: Yes

5. Is the manuscript presented in an intelligible fashion and written in standard English?

Reviewer #1: Yes

Reviewer #2: Yes

6. Review Comments to the Author

Reviewer #1: (No Response)

Reviewer #2: I did think the authors have made a proper revision upon the reviewers' comments, and this paper is qualified to be published.

7. PLOS authors have the option to publish the peer review history of their article (what does this mean?). If published, this will include your full peer review and any attached files.

Reviewer #1: No

Reviewer #2: No

---

## [Editor Report · Acceptance letter]

4 Feb 2022

PONE-D-21-35527R1 

A comparative driving safety study of mountainous expressway individual tunnel and tunnel group based on Eye Gaze Behavior 

Dear Dr. Huang:

I'm pleased to inform you that your manuscript has been deemed suitable for publication in PLOS ONE. Congratulations! Your manuscript is now with our production department. 

Kind regards, 

on behalf of

Dr. Feng Chen 

Academic Editor

PLOS ONE